# Potential Role of miRNAs in the Acquisition of Chemoresistance in Neuroblastoma

**DOI:** 10.3390/jpm11020107

**Published:** 2021-02-07

**Authors:** Barbara Marengo, Alessandra Pulliero, Maria Valeria Corrias, Riccardo Leardi, Emanuele Farinini, Gilberto Fronza, Paola Menichini, Paola Monti, Lorenzo Monteleone, Giulia Elda Valenti, Andrea Speciale, Patrizia Perri, Francesca Madia, Alberto Izzotti, Cinzia Domenicotti

**Affiliations:** 1Department of Experimental Medicine, University of Genova, 16100 Genova, Italy; lolloleo92@gmail.com (L.M.); giuliaelda.valenti@edu.unige.it (G.E.V.); alberto.izzotti@unige.it (A.I.); cinzia.domenicotti@unige.it (C.D.); 2Department of Health Sciences, University of Genova, 16100 Genova, Italy; alessandra.pulliero@unige.it; 3Laboratory of Experimental Therapies in Oncology, IRCCS Istituto Giannina Gaslini, 16100 Genova, Italy; mariavaleriacorrias@gaslini.org (M.V.C.); patriziaperri@gaslini.org (P.P.); 4Department of Pharmacy, University of Genova, 16100 Genova, Italy; riclea@difar.unige.it (R.L.); farinini@difar.unige.it (E.F.); 5UOC Mutagenesis and Cancer Prevention, IRCCS Ospedale Policlinico San Martino, 16100 Genova, Italy; gilberto.fronza@hsanmartino.it (G.F.); paola.menichini@hsanmartino.it (P.M.); paola.monti@hsanmartino.it (P.M.); andreaspeciale@alice.it (A.S.); 6Medical Genetics Unit, IRCCS Giannina Gaslini Institute, 16100 Genova, Italy; francescamadia@gaslini.org

**Keywords:** neuroblastoma, miRNA, *MYCN* amplification, metastases, chemoresistance

## Abstract

Neuroblastoma (NB) accounts for about 8–10% of pediatric cancers, and the main causes of death are the presence of metastases and the acquisition of chemoresistance. Metastatic NB is characterized by *MYCN* amplification that correlates with changes in the expression of miRNAs, which are small non-coding RNA sequences, playing a crucial role in NB development and chemoresistance. In the present study, miRNA expression was analyzed in two human *MYCN*-amplified NB cell lines, one sensitive (HTLA-230) and one resistant to Etoposide (ER-HTLA), by microarray and RT-qPCR techniques. These analyses showed that miRNA-15a, -16-1, -19b, -218, and -338 were down-regulated in ER-HTLA cells. In order to validate the presence of this down-regulation in vivo, the expression of these miRNAs was analyzed in primary tumors, metastases, and bone marrow of therapy responder and non-responder pediatric patients. Principal component analysis data showed that the expression of miRNA-19b, -218, and -338 influenced metastases, and that the expression levels of all miRNAs analyzed were higher in therapy responders in respect to non-responders. Collectively, these findings suggest that these miRNAs might be involved in the regulation of the drug response, and could be employed for therapeutic purposes.

## 1. Introduction

Neuroblastoma (NB) is an extracranial pediatric tumor originating from the aberrant development of neural crest-derived sympathoadrenal lineage [1], and is characterized by a high clinical and biological heterogeneity [2]. In fact, NB can be classified as a low-risk tumor, capable of spontaneously regressing, as well as a high-risk tumor, responsible for a high mortality rate and characterized by the presence of metastases.

The therapy used in high-risk patients is multimodal, and although the response to treatments is initially positive, subsequently, following the onset of chemoresistance, a large number of patients die as a consequence of relapse and metastasis formation [3]. NB metastasizes in vascularized tissue, and bone marrow (BM) is the preferential site of recurrence, being considered as the “fertile soil” for tumor cells and, in particular, for the chemoresistant cells [4,5]. In fact, BM spread is considered a negative prognostic factor [6].

Among the prognostic markers of poor patient outcome, the amplification of the *MYCN* oncogene characterizes the most aggressive high-risk NB subtype [7,8]. More than ten years ago, it was proposed that another approach to classifying the risk group of NB patients could be to evaluate the expression levels of miRNAs [9]. Considering that *MYCN* modulates the expression of several miRNAs [10], the evaluation of these small non-coding RNA sequences has become an accurate predictor of NB outcome [11].

Furthermore, since miRNA expression is related to tumor grade, metastasis, and chemoresistance, they could represent a new class of potential therapeutic targets. In this context, we have recently demonstrated in an NB cell line-based model [12] that Etoposide resistance is associated with miRNA-15a/16-1 down-regulation, highlighting that miRNAs could have a role as both markers of chemoresistance and new possible therapeutic targets [13]. Therefore, in the present study, the expression of these miRNAs, amongst others, was analyzed in primary tumors, metastases, and bone marrow of therapy responder and non-responder NB patients in order to identify the specific miRNAs involved in NB progression and chemoresistance.

## 2. Materials and Methods

### 2.1. Cell Cultures

The *MYCN*-amplified human stage-IV NB cell line, HTLA-230, was obtained from Dr. L. Raffaghello (G. Gaslini Institute, Genoa, Italy), while the Etoposide-resistant cell line (ER-HTLA) was selected as previously reported [12,13]. Cells were periodically tested for mycoplasma contamination (Mycoplasma Reagent Set, Aurogene s.p.a, Pavia, Italy). Cells were cultured in RPMI 1640 (Euroclone SpA, Pavia, Italy) and supplemented with 10% fetal bovine serum (FBS; Euroclone SpA, Pavia, Italy), 2 mM of glutamine (Euroclone SpA, Pavia, Italy), 1% penicillin/streptomycin (Euroclone SpA, Pavia, Italy), 1% sodium pyruvate (Sigma-Aldrich, Saint Louis, MO, USA), and 1% amino acid solution (Sigma-Aldrich, Saint Louis, MO, USA).

### 2.2. Patient Samples

The patients included in the study were diagnosed with NB stage M between January 2002 and December 2015. Written consent for the use of samples and clinical data for research was obtained by their legal guardians. The study was approved by the Gaslini Institute Ethical Committee, and all analyses were performed according to the Helsinki declaration.

The samples used originated from two groups of patients. With regard to the first group, whole bone marrow (BM) samples were collected in PAXgene™ Blood RNA tubes originating from patients who, after diagnosis, were treated according to the high-risk European protocol. The drugs used in the induction therapy were Cisplatin, Etoposide, Vincristine, Cyclophosphamide, and either Carboplatin or Adriamycin. Patients were divided into two subgroups, responders and non-responders: responders being the patients that could proceed with high-dose chemotherapy and stem cell transplants, and non-responders being the patients who could not proceed and were referred to second-line therapy.

With regards to the second group, the samples were represented by tumor specimens containing more than 70% of neoplastic cells and immune-selected metastases from BM samples, as described [14,15] and containing 95% tumor cells. All patient samples were taken at diagnosis before starting with the treatment.

### 2.3. RNA Extraction

Total RNA was extracted from cultured cells using TRIZOL reagent (LifeTechnologies, Carlsbad, CA, USA) according to the manufacturer’s instructions. Total RNA (1 μg) was reverse-transcribed into cDNA by a random hexamer primer and SuperScript™ II Reverse Transcriptase (LifeTechnologies, Carlsbad, CA, USA).

Total RNA and miRNA fractions were extracted from tumor cells and metastases using the miRNeasyMini kit (Qiagen, Hilden, Germany), according to the manufacturer’s protocols. Total RNA and miRNA fractions were extracted from whole BM samples using the PAXgene extraction Kit (Qiagen, Hilden, Germany), according to manufacturer’s protocol. The quality of the RNA fractions was evaluated in the BioAnalyzer 2100 system (Agilent Technologies, Santa Clara, CA, USA).

### 2.4. MiRNA Microarray Analysis 

MiRNA expression profiling was carried out by the Agilent platform following the miRNA Microarray protocol v.3.1.1 (Agilent Technologies, Santa Clara, CA, USA). Briefly, 50 ng of total RNA, containing miRNAs and spike-in controls, underwent dephosphorylation and a labeling step with Cyanine 3-pCp. The Cy3-labeled RNA was then purified using the Micro Bio-Spin P-6 Gel Column (Bio-Rad Laboratories, Inc., Hercules, CA, USA), and hybridized on human miRNA microarray slides 8×60K (Agilent Technologies; including 2549 miRNAs, miRBase 21.0) at 55 °C for 20 hours. After washing, the slides were scanned by a G2565CA scanner (Agilent Technologies, Santa Clara, CA, USA), and the images were extracted by Feature Extraction software v.10 (Agilent Technologies, Santa Clara, CA, USA). Tab-delimited text files were analyzed in R v.2.7.2 software environment http://www.r-project.org using the limma package v.2.14.16 of Bioconductor http://www.bioconductor.org. Only spots with a signal minus background flagged as positive and significant were used in the following analysis as detected spots. Probes with less than 50% of detected spots across all arrays and arrays with a number of detected spots smaller than 50% of all spots on the array were removed. Background corrected intensities of replicated spots on each array were averaged. Data were then log2-transformed and normalized for between-array comparison using quantile normalization [16]. MicroRNAs with *p*-values < 0.05 were selected for further analysis. Given the explorative nature of this study, no correction for multiple testing was applied to the screening procedure aimed at selecting multiple sets of microRNAs for subsequent hierarchical clustering analyses. The agglomerative hierarchical clusters, used to detect similarity relationships in microRNA log2-transformed expressions, were computed by the Euclidean distance between single vectors and the Ward method [17].

### 2.5. Real Time PCR Analysis 

Total RNA (10 ng) was reverse transcribed using miR-specific stem-loop RT primers (TaqMan MicroRNA Assays; Applied Biosystems, Thermo-Fisher, Waltam, MA, USA) and components of the High Capacity cDNA Reverse Transcription kit (Life Technologies, Carlsbad, CA, USA), according to the manufacturer’s protocols. Expression levels of individual miRNAs were detected by subsequent RQ-PCR using TaqMan MicroRNA assays (Life Technologies, Carlsbad, CA, USA) and a Rotor Gene 3000 PCR System Corbett (Qiagen, Hilden, Germany) with standard thermal cycling conditions, in accordance with manufacturer recommendations. PCR reactions were performed in triplicate in final volumes of 30 µl, including inter-assay controls (IAC) to account for variations between runs. RT-PCR (TaqMan MicroRNA Assays; Applied Biosystems, Thermo-Fisher) was used to quantify the expression of has-miR-16, has-miR-15a, has-miR-19b, has-miR-26b, has-miR-27b, has-miR-29c, has-miR-34c, has-miR-126, has-miR-218, has-miR-338, and has-miR-497, according to the manufacturer’s instructions. To normalize the data for quantifying miRNAs, the universal small nuclear RNU38B (RNU38B Assay ID 001004; Applied Biosystems, Thermo-Fisher, Waltam, MA, USA) as an endogenous control was used [18].

The delta–delta Ct method was employed to calculate the fold change. In brief, each 15 μL of the reaction system contained 0.15 μL of 100 mM dNTPs with dTTP, 1 μL of MultiScribe Reverse Transcriptase (50 U/μL), 1.5 μL of RT buffer (×10), 0.1  μL of RNase inhibitor (20 U/μL), 6.25  μL of nuclease-free water, 5  μL of small RNA, and 3  μL of RT primer. Small RNAs were quantified by a Qubit 3 fluorimeter (Life Technologies, Carlsbad, CA, USA). Thermal cycling conditions were 30 min at 16 °C, 30 min at 42 °C, and 5 min at 85 °C. Each 20 μL of the reaction system for real-time quantitative PCR contained 1 μL of real-time primer, 1.33  μL of product from the RT reaction, 10  μL of TaqMan Universal PCR Master Mix, and 7.67  μL of nuclease-free water. The reactions were performed in triplicate on a Rotor Gene 3000 PCR System Corbett for 10 min at 95 °C, followed by 40 cycles of 15 s at 95 °C and 1 min at 60 °C. Along with the Cq values calculated automatically by the SDS software (threshold value = 0.2, baseline setting: cycles 3–15), raw fluorescence data (Rn values) were exported for further analyses.

### 2.6. Comparative Genomic Hybridization (CGH) Analysis 

Array CGH analyses were performed using the Human Genome array-CGH 8 × 60 K Microarray (Agilent Technologies, Palo Alto, CA), with an average probe spacing of around 55 Kb.

The arrays were performed using Agilent Reference DNAs, analyzed with the Agilent Microarray Scanner Feature Extraction Software version 11.5, and Agilent Genomic Workbench 7.0.4.0 software using the ADM-2 algorithm. Genomic positions of the rearrangements refer to the public UCSC database GRCh37.

### 2.7. PCA Analysis 

Principal components analysis (PCA) is a data display method for multivariate data. 

Given a data set in which each sample is described by *n* variables, the PCA aims to find new directions and linear combinations of the original ones [19,20].

The first component (PC1) corresponds to the direction explaining the maximum variance, while PC2 is the direction, orthogonal to PC1, explaining the maximum variance not explained by PC1, and so on. The result of such a transformation is that a limited number of components is sufficient to explain the relevant part of the information.

The loadings are the coefficients of the linear combinations corresponding to the PCs. By plotting them in a loading plot, it is possible to understand the relationships among the variables in the multivariate space. 

On the other side, the score plot (the scores being the coordinates of the samples in the new space defined by the PCs) allows the visualization of the location of samples in the space described by the PCs, making it possible to check similarities and differences among the samples. 

The elaborations and the plots were carried out through the software CAT (Chemometric Agile Tool, www.gruppochemiometria.it) [21].

### 2.8. Statistical Analysis

Results were expressed as mean ± SEM from at least three independent experiments. The statistical significance of the parametric differences among the sets of experimental data was evaluated by one-way ANOVA and Dunnett’s test for multiple comparisons. Statistical analysis of the mitotic index and reporter assays data was performed using the Fisher’s exact test.

## 3. Results and Discussion

### 3.1. miRNA Expression Profiling of HTLA-230 and ER-HTLA Cells. 

In order to identify the miRNAs involved in chemoresistance, miRNA microarray analyses were performed on HTLA-230 and ER-HTLA cells.

As shown in Figure 1, miRNAs were differently expressed when comparing these two cell populations. The scatter plot analysis showed that a total of 152 miRNAs changed their expression more than 1.5 fold, 41 being up-regulated and 111 down-regulated (Figure 1). 

Volcano plot analyses, considering threshold values of five-fold for fold variation and *p* < 0.01 for statistical significance, showed that a total of 35 miRNAs significantly changed their expression, three being up-regulated and 32 down-regulated. The list of these 35 miRNAs is available in the Supplementary Material (Appendix A).

Given the mechanism of action of miRNAs in regulating gene expression, we focused our attention on the down-regulated ones, and, in order to restrict the number of miRNAs to be studied, from the literature we searched for miRNAs that had been specifically involved in NB biology and/or chemoresistance. Using this criterion, 11 miRNAs were selected (Table 1), and their expression was tested by RT-qPCR analysis. 

As shown in Figure 2, only six miRNAs (i.e., miR-15a, -16-1, -19b, -27b, -126, and -218) among the selected miRNAs were confirmed to be down-regulated in ER-HTLA cells in respect to HTLA-230 parental ones. In detail, miR-27b and miR-16-1 expression levels were found to be reduced by 33.1 and 23.5 fold, respectively, and miR-218 expression was diminished by 9.09 fold, while miR-15a, miR-126, and miR-19b were down-regulated (slightly, but significantly) by 2.8, 2.7, and 1.73 fold, respectively. 

For the first time, to our knowledge, this data confers a possible role in NB chemoresistance to miR-27 and miR-218. In fact, despite their expression being found to be down-regulated in several chemoresistant cancers (see Table 1) [48,49,50,71,72,73,74,75], their involvement in chemoresistance of NB has never been reported in the literature. Notably, although miRNA-218 was found to be up-regulated in *MYCN*-amplified and metastatic NB [68,69,70], this data is not in contradiction with the down-regulation of miRNA-218 that we have observed after chronic Etoposide exposure of *MYCN*-amplified NB cells (ER-HTLA).

In addition, these results confirm the down-regulation of miR-15 and miR-16 in ER-HTLA cells, as found in our previous study [13]. Moreover, for the first time, miR-19b expression was found to be reduced in chemoresistant NB in conformity with other malignancies (Table 1) [38,39,40]. In fact, only one study has reported an up-regulation of miR-19b in chemoresistant NB cells [37]. This discrepancy could be due to the fact that, in this same study, NB cells were exposed to the drug for only 24 hours while, in our present study, the ER-HTLA cells were selected by chronically treating parental cells (HTLA-230) with Etoposide for six months (i.e., a condition that better mimics in vivo treatment). 

### 3.2. Comparative Genomic Hybridization (CGH) on HTLA-230 and ER-HTLA Cells

Since both genetic and epigenetic mechanisms have been demonstrated to influence NB biology [86], in order to better characterize the chemoresistant phenotype, CGH analysis was performed. DNAs from HTLA-230 and ER-HTLA cells were hybridized to obtain a comparison of gains and losses that could be connected to the acquisition of chemoresistance. As reported in Table 2, an intriguing finding was the presence of several alterations in chromosome 13, where miR-15a, miR-16-1, and miR-19b were mapped, and, in chromosome 17, where miR-338 was localized.

Moreover, our data, identifying some chromosomal regions that are more frequently altered in ER-HTLA cells, is in line with the results obtained in a previous paper reporting a gain of 13q14.1-32 and a loss of 17q in other NB chemoresistant cell lines [87]. Since these chromosome traits (e.g., chromosome 13) contain the locus in which miRNAs, involved in the acquisition of Etoposide resistance, are mapped, it is possible to hypothesize that the evaluation of these miRNAs in patient samples might be used as prognostic markers that are able to early identify chemoresistant signatures. 

### 3.3. miRNA Expression Profiling of Therapy-Sensitive (Responder) and Therapy-Resistant (Non-Responder) NB Patients

In order to evaluate in vivo the expression of miRNAs and their potential role in NB chemoresistance, ten whole BM samples, taken at diagnosis from NB patients either sensitive (responder) or resistant (non-responder) to induction therapy, were randomly selected from our biobank, and six miRNAs from Table 1 (i.e., miR-15a, -16-1, 19b, -27b, -126, and -218) were analyzed. NB patient characteristics are reported in Table 3.

By comparing the expression of miRNAs in therapy-sensitive and therapy-resistant NB patients, only miR-16 was significantly down-regulated in the bone marrow of non-responder patients (Figure 3), while the other miRNAs analyzed were not significantly modified, even though a slight trend of reduction in non-responder patients was observed.

These findings, while confirming a potential role of miR-16 in delivering intrinsic chemoresistance of NB, do not confirm the other results obtained from ER-HTLA cells. However, this is not unusual when comparing in vitro with in vivo data, most likely due to the variability found in individual patients. Nevertheless, it should be noted that the content of NB cells in the BM samples ranged from 5% to 35%, making the normal hematopoietic cells prevalent, and thus potentially masking miRNA down-regulation occurring in neoplastic cells.

### 3.4. miRNA Expression Profiling of NB Primary Tumors and Metastases

Therefore, in order to better understand the role that these miRNAs could possibly have in NB biology, their expression was tested in ten primary tumors and ten immune-magnetically-enriched NB metastases from stage M NB patients, randomly selected from our biobank. The NB patients’ features of this new set of samples are reported in Table 4. Since it has been recently reported [13] and herein confirmed that ER-HTLA cells have a monoallelic deletion of the 13q14.3 locus, which maps for miR-15/16, particular attention was given to those miRNAs whose locus was found mutated. The analysis was also extended to miR-338 and miR-218, even though the corresponding locus had not been altered, because their expression has been demonstrated to be strictly related to NB chemoresistance [77,78] and to Etoposide refractoriness [76]. 

As reported in Figure 4, miRNA-19b, -218, and -338 were down-regulated in *MYCN*-amplified metastases by about 30% as compared to *MYCN*-amplified tumors, while no significant changes were observed in the expression of the other miRNAs. It is interesting to note that the *MYCN* status did not influence the expression of these latter miRNAs, neither in tumors nor in metastases (Figure 4).

miR-338 down-regulation in NB metastases has been previously reported by Chen et al., who demonstrated that this miRNA could exert an inhibitory role on the migration, proliferation, and invasion of NB cells through the modulation of the PTEN/Akt pathway [77]. However, while miR-19b expression has been found to be reduced in metastatic clear renal cell carcinoma [88] and miR-218 expression down-regulated in metastatic prostate [89], breast [90], gastric [91], cervical [92], and lung [93] cancer, to our knowledge, this is the first time that a down-regulation of miR-19b has been detected in metastatic NB in vivo. In addition, miRNA-218 has also been found to be down-regulated in NB metastases. This result obtained from analyses of patients’ tissues is in contrast with previous studies reporting an up-regulation of this miRNA [70] in patients’ serum, but this discrepancy could be due to the different nature of the analyzed samples. In fact, it is conceivable that the increased levels of miRNA-218 in the serum might be due to a response of peritumoral tissue, and not originating from the tumor. 

### 3.5. Principal Component Analysis (PCA) of the Results Obtained in Patients’ Samples

In order to better extract the information from the dataset about NB biology and chemoresistance, principal component analysis (PCA) was carried out by collecting miRNA expression profiles analyzed in bone marrows infiltrates, tumors, and metastases. The first PCA has been performed on responder (four) and non-responder (four) patients’ samples. The loading plot showed that all of the variables had similar positive loadings on PC1, meaning that the score on PC1 can be considered as a global quantitative index. On the other hand, PC2 mainly explains the contrast between miR-15, miR-218, and -16 variables (Figure 5, left panel). By analyzing the score plot, although only a few subjects were available, it was possible to observe that all of the responder patients’ samples (red) were well-separated from the non-responders (black) and located in a specific region in the PC space: compared to the non-responders, all of the responders were mainly characterized by higher values of the variables miR-218 and miR-16 (Figure 5, right panel).

Furthermore, a second PCA was carried out on tumor and metastasized patients’ samples. The loading plot showed correlations between the variables miR-19b, -218, and -338, all characterized by negative loadings on PC1 (group 1), and between miR-15 and -16, which had positive loadings on PC2 (group 2). The two groups of correlated variables were uncorrelated, since their directions from the origin were orthogonal (Figure 6, left panel). The score plot showed that the metastasized patients’ samples were characterized by intermediate values of the variables miR-15 and -16. On the other hand, the majority of the non-metastasized patients’ samples had extreme values of both variables miR-15 and -16 (Figure 6, right panel).

## 4. Conclusions

The presence of chemoresistant cells in the primary tumor and within bone marrow is the most powerful negative prognostic factor for patients with NB. The acquisition of chemoresistance and the ability to metastasize are the results of genetic and epigenetic mechanisms and, among them, miRNAs can play a crucial role [94]. In fact, their expression is frequently de-regulated in several chemoresistant malignancies and, as supported by the results herein, in NB. Indeed, the evaluation of miRNA expression could have a double value. In fact, on the one hand, the modulation of a specific miRNA or of a cluster of miRNAs could be used as a prognostic and predictive factor advantageous for monitoring the acquisition of chemoresistance. On the other hand, miRNAs might also have therapeutic potential, since many current studies are focused on discovering the best mechanism that is able to restore the expression of miRNAs in oncologic patients [95]. In the present study, our findings demonstrate, for the first time, that the down-regulation of miR-16-1 is strictly related to the acquisition of NB chemoresistance. In fact, among the six miRNAs whose expression is found down-regulated in our in vitro model of Etoposide resistance, only miR-16-1 is significantly down-regulated in non-responder NB patients treated with the induction therapy comprised of Etoposide (Figure 7). This data suggests that the restoration of miR-16 could be a valid strategy to counteract chemoresistance (Figure 7).

In addition, miR-19b, miR-338, and partially miR-218, whose down-regulation correlates with the metastatic process, could have prognostic value as biomarkers of NB progression. 

## Figures and Tables

**Figure 1 jpm-11-00107-f001:**
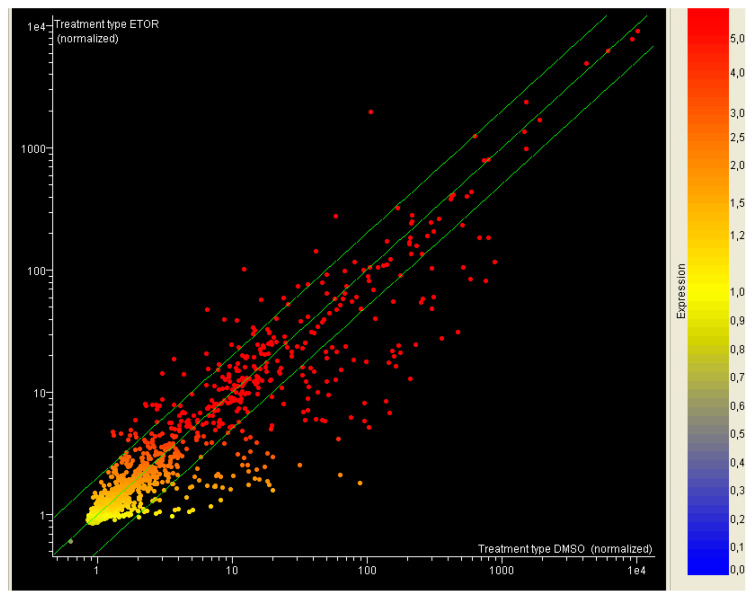
Scatter plot analysis reporting the variation of miRNA expression between HTLA-230 (horizontal axis) and ER-HTLA cells (vertical axis). Each dot represents one miRNA colored according to its level of expression. Green diagonal lines indicate the 1.5-fold variation interval.

**Figure 2 jpm-11-00107-f002:**
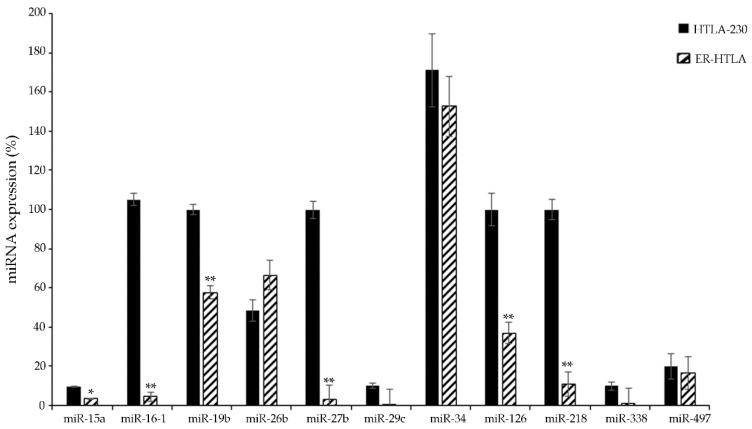
Evaluation of the expression level of the 11 selected miRNAs (Table 1) in HTLA-230 and ER-HTLA cell lines by RT-qPCR analysis. Data is reported as % variation vs. the universal small nuclear RNU38B. * *p* < 0.05 vs. HTLA-230 ** *p* < 0.01 vs. HTLA-230.

**Figure 3 jpm-11-00107-f003:**
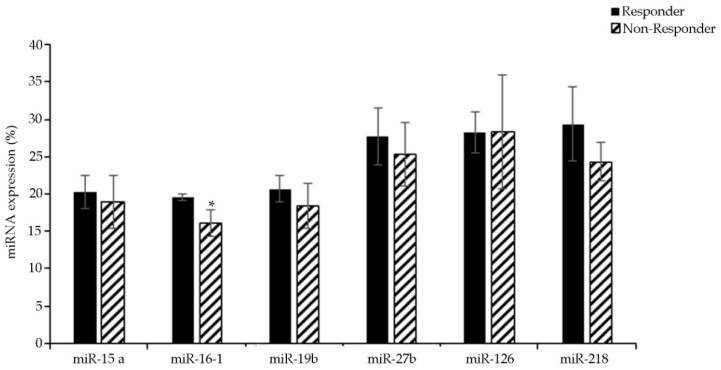
Evaluation of the expression levels of selected miRNAs in the bone marrow of patients sensitive (responder) or resistant (non-responder) to therapy by RT-qPCR analysis. Data is reported as % variation vs. the universal small nuclear RNU38B. * *p* < 0.05 vs. responder.

**Figure 4 jpm-11-00107-f004:**
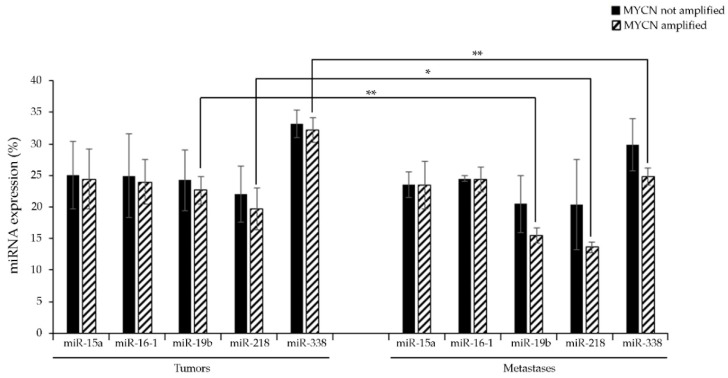
RT-qPCR analysis of the selected miRNA expression levels in tumors and metastases samples of NB patients. Data is reported as % variation vs. the universal small nuclear RNU38B. * *p* < 0.05 vs. *MYCN*-amplified tumors; ** *p* < 0.01 vs. *MYCN*-amplified tumors.

**Figure 5 jpm-11-00107-f005:**
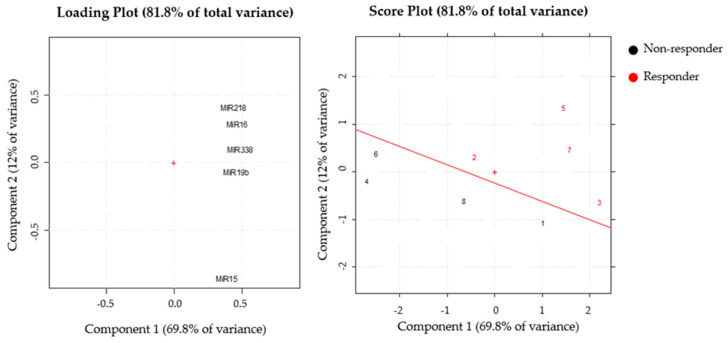
PCA performed on responder and non-responder patients’ samples. The loading (left panel) and the score plot (right panel) were reported. In the score plot, the samples were indicated by the number reported in Table 3. + represents the point with coordinates 0 and 0 for x and y axes, respectively. This points is the reference to define the multivariate space.

**Figure 6 jpm-11-00107-f006:**
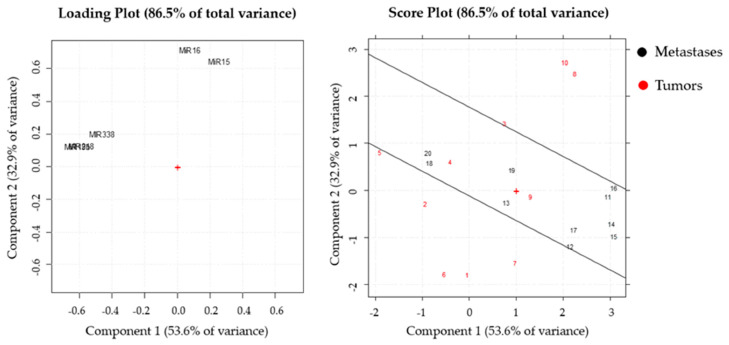
PCA performed on tumors and metastases samples. The loading (left panel) and the score plot (right panel) were reported. In the score plot, the samples were indicated by the number reported in Table 4. + represents the point with coordinates 0 and 0 for x and y axes, respectively. This points is the reference to define the multivariate space.

**Figure 7 jpm-11-00107-f007:**
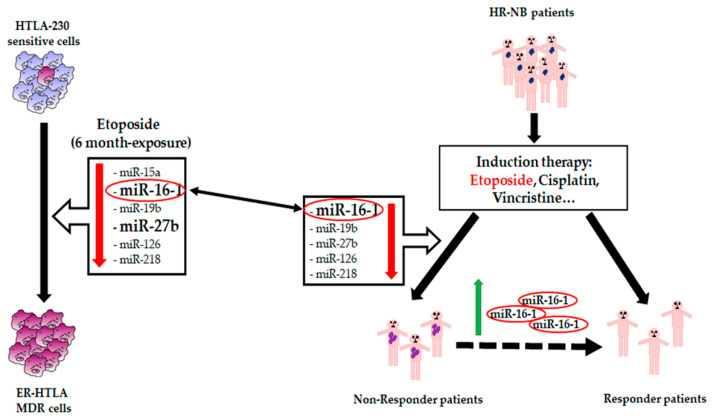
Role of miR-16-1 in NB chemoresistance. The acquisition of chemoresistance in multidrug resistant (MDR) cells and in high-risk (HR) patients is characterized by miR-16-1 down-regulation, suggesting a potential role of miR-16-1 as a chemosensitizer.

**Table 1 jpm-11-00107-t001:** miRNAs differently expressed in HTLA-230 and ER-HTLA cell lines that are involved in NB biology and in general cancer chemoresistance.

miRNA	HTLA-230/ER-HTLA Ratio	Expression in Neuroblastoma	Expression in Other Chemoresistant Cancers
miR-15a	11.38	Down-regulated in *MYCN*-amplified chemoresistant NB [13,22,23,24]	Down-regulated in Burkitt Lymphoma [25], pancreatic ductal adenocarcinoma [26], colorectal [27], and ovarian cancer [28]
miR-16	7.89	Down-regulated in *MYCN*-amplified NB [22,29] and in chemoresistant NB [13]	Down-regulated in cervical [30], breast [31,32] gastric [32,33], and lung [34] cancer, osteosarcoma [35], and mesothelioma [36]
miR-19b	7.75	Up-regulated in chemoresistant NB [37]	Down-regulated in breast [38] and colon [39] cancer and leukemia [40]
miR-26b	47.87	Not evaluated	Down-regulated in chemoresistant colorectal [41], gastric [42], laryngeal [43], and hepatocellular carcinoma [44,45] cancer and in glioma [46]
miR-27b	8.29	Down-regulated in NB [47]	Down-regulated in lung [48], breast [49], and gastric cancer [50]
miR-29c	9.27	Not evaluated	Down-regulated in ovarian [51], endometrial [52], gastric [53], and small cell lung [54] cancer, glioma [55,56], and leukemia [57,58]
miR-34c	7.49	Not evaluated	Down-regulated in colon [59], gastric [60,61], and ovarian [62,63] cancer, and osteosarcoma [64]
miR-126a	9.66	Not evaluated	Down-regulated in colorectal [65] and breast cancer [66] and in renal cell carcinoma [67]
miR-218	12.30	Up-regulated in *MYCN*-amplified and in metastatic NB [68,69,70]	Down-regulated in glioma cells [71], colorectal [72], gallbladder [73], bladder [74], and lung cancer [75,76]
miR-338	15.99	Down-regulated in resistant NB [77,78]	Down-regulated in esophageal squamous carcinoma cells [79]
miR-497	6.92	Down-regulated in chemoresistant NB [24], in *MYCN*-amplified NB [80]	Down-regulated in lung [81], colorectal [82], ovarian [83], and pancreatic [84] cancer, and lymphoma [85]

**Table 2 jpm-11-00107-t002:** CGH analysis on HTLA-ER cells in comparison to HTLA-230 cells.

CHR	START	STOP	CYTO	SIZE KB	VALUE	Control (HTLA-230)
**1**	152,079,488	155,154,990	q21.3–q22	3.075	1.5	Absent
**3**	73,792,065	75,028,724	p13–p12.3	1.236	−0.7	Absent
**5**	20,160,410	44,924,503	p14.3–p12	24.764	−0.7	Absent
**8**	112,697,432	146,280,020	q23.3–q24.3	33.582	−1/−0.4	Duplicated
**9**	204,193	38,815,475	p24.3–p13.1	38.611	−0.7/−3	Absent
**10**	43,020,732	60,914,512	q11.21–q21.1	17.893	−0.7/−1.2	Duplicated
**12**	38,805,636	48,103,580	q12–q13.11	9.297	0.7/0.4/1.4	Absent
**13**	20,412,619	39,841,779	q12.11–q13.3	19.429	0.3/0.6	Absent
**13**	39,900,189	86,110,407	q13.3–q31.1	46.210	−0.5	Absent
**13**	86,151,801	111,106,213	q31.1–q34	24.954	0.5	Absent
**13**	111,181,035	113,538,619	q34	2.357	−0.8	Absent
**13**	113,610,612	115,092,648	q34	1.482	0.4	Absent
**17**	44,684	625,475	p13.3	580	−0.7	Absent
**17**	25,654,874	40,109,636	q11.1–q21.2	14.454	−0.7/−1.2	Duplicated
**19**	32,783,771	36,293,337	q13.11–q13.12	3.509	−0.6	Duplicated
**20**	60,747	19,483,849	p13–p11.23	19.423	0.5	Deleted or mosaic
**21**	15,538,980	32,776,404	q11.2–q22.11	17.237	−0.6	Absent

**Table 3 jpm-11-00107-t003:** NB characteristics and patient clinical outcomes.

N	MYCN Status	Age (Months)	EFS (Months)	OS (Months)	INRG Stage	Induction Response	Relapse	Follow-Up
2	Amplified	55	71.25	71.25	M	Yes	No	Alive
3	Not evaluated	41	81.06	81.06	M	Yes	No	Alive
4	Single copy	12	60.53	60.53	M	Yes	No	Alive
6	Amplified	62	50.83	50.83	M	Yes	No	Alive
9	Amplified	21	55.38	55.38	M	Yes	No	Alive
1	Amplified	17	6.86	7.10	M	No	Yes	Dead
5	Single copy	47	22.94	26.17	M	No	Yes	Dead
7	Amplified	20	9.54	19.27	M	No	Yes	Dead
8	Amplified	21	5.54	7.00	M	No	Yes	Dead
10	Amplified	16	4.69	8.98	M	No	Yes	Dead

EFS, event-free survival; OS, overall survival; INRG, International Neuroblastoma Risk Group.

**Table 4 jpm-11-00107-t004:** Features of NB primary tumors and metastases and patients’ clinical outcomes.

	N	MYCN Status	Age (Years)	EFS (Months)	OS (Months)
Tumors	1	Amplified	1.99	46.2	84.8
	2	Not amplified	3.18	5.3	9.3
	3	Not amplified	1.27	187.2	187.2
	4	Amplified	1.13	4.3	7.4
	5	Amplified	3.88	70.0	114.5
	6	Amplified	6.30	36.2	47.7
	7	Amplified	2.07	3.2	10.3
	8	Amplified	4.76	114,9	114,9
	9	Amplified	4.57	21.7	22.4
	10	Amplified	1.44	26.8	33.0
Metastases	11	Not amplified	1.68	58.88	58.88
	12	Not amplified	3	24.52	35.98
	13	Amplified	6.8	8.68	42.41
	14	Amplified	0.9	11.06	11.06
	15	Not amplified	2.55	70.73	70.73
	16	Not amplified	3.34	13,3	21.22
	17	Amplified	8.2	8.61	8.61
	18	Amplified	1.67	9.6	14.98
	19	Amplified	6.89	29.04	29.04
	20	Amplified	1.7	13.3	23

EFS, event-free survival; OS, overall survival.

## Data Availability

Data are available upon request from the corresponding author. Data are not publicly available due to patients’ privacy.

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
