# Peer review of "Potential Role of miRNAs in the Acquisition of Chemoresistance in Neuroblastoma"

_jpm, 2021, doi:10.3390/jpm11020107_

Round 1

Reviewer 1 Report

The objective of this study was to identify miRNAs associated with the acquisition of metastatic potential and chemoresistance in neuroblastoma. By comparing the miRNA profile of an MYCN-amplified human stage-IV NB cell line (HTLA-230) with that of an Etoposide-resistant cell line (ER-HTLA), it appeared that miRNA-15a, -16-1, -19b, -218 and -338 was downregulated in ER-HTLA neuroblastoma cells. Those miRNAs were found to express at high levels in the therapy-responders than in the non-responders, suggesting the identified miRNAs are involved in involved in the regulation of drug response to chemotherapeutic agents in neuroblastoma. 

Major points:

Line 32-33: "These analyses showed that miRNA-15a, -16-1, -19b, -218 and -338 down-regulation correlated with the acquisition of chemoresistance". Where were the data showing downregulation of the expression of those miRNAs were correlated with acquisition of chemoresistance? What correlation test was used?

Line 81, subsection 2.2. How many patients were recruited in this study? There were 10 patients in Table 3, while 20 patients were listed in Table4. Were those two different groups of patients?

Line 87 - 88. Patients recruited to this study were given cisplatin, vp-16, vincristine, cyclophosphamide, and either carboplatin or Adriamycin. Etoposide was not one of the chemotherapeutic agents given to those patients. Why was Etoposide-resistant NB cell line used to identify miRNAs associated with chemoresistance to other chemo drugs? Have there been any published studies indicating that an NB cell line resistant to different cytotoxic chemotherapeutic agents would have the same miRNA expression profile? Also, chemoresistance can be intrinsic or acquired. Did those non-responders exhibit intrinsic resistance or acquired resistance to the treatment?

Regarding the data shown in Table 2 and Table3, survival analysis should be conducted, and Kaplan-Meier survival curves for OS and PFS should be presented.

Author Response

Genova, 20th January 2021

For the kind attention of Editor-in-Chief

Journal of Personalized Medicine

Dear Editor,

Firstly, we would like to thank you for having reviewed the paper and thank the Reviewers for their most welcome and helpful comments that have gone towards improving the quality of the manuscript which is now enclosed.

In this revision phase, we have tried as far as possible to satisfy the requests and comments by modifying the main text.

For greater clarity, the changes made to the original manuscript have been highlighted in the text as follows:

-           parts added in red;

-           parts removed in red but barred.

As you will see, we have addressed all the points raised by the Reviewers and amended the manuscript accordingly as follows:

REVIEWER 1

Comments and Suggestions for Authors

The objective of this study was to identify miRNAs associated with the acquisition of metastatic potential and chemoresistance in neuroblastoma. By comparing the miRNA profile of an MYCN-amplified human stage-IV NB cell line (HTLA-230) with that of an Etoposide-resistant cell line (ER-HTLA), it appeared that miRNA-15a, -16-1, -19b, -218 and -338 was downregulated in ER-HTLA neuroblastoma cells. Those miRNAs were found to express at high levels in the therapy-responders than in the non-responders, suggesting the identified miRNAs are involved in involved in the regulation of drug response to chemotherapeutic agents in neuroblastoma.

Major points:

  • Line 32-33: "These analyses showed that miRNA-15a, -16-1, -19b, -218 and -338 down-regulation correlated with the acquisition of chemoresistance". Where were the data showing downregulation of the expression of those miRNAs were correlated with acquisition of chemoresistance? What correlation test was used?

In agreement with the Reviewer's comment, we have modified the sentence in the abstract better explaining that miRNA-15a, -16-1, -19b, -218 and -338 were found to be down-regulated in Etoposide-resistant NB cells (lines 33-34).

  • Line 81, subsection 2.2. How many patients were recruited in this study? There were 10 patients in Table 3, while 20 patients were listed in Table4. Were those two different groups of patients?

We are sorry for the misunderstanding. We confirm that the groups of patients were actually two. With regard to the first group, the unselected bone marrow samples were analyzed. With regard to the second one, primary tumors and metastases were analyzed. In agreement with the Reviewer’s concern, we have better explained this point in the Materials and Methods section (lines 85-91) and in the Results section (lines 284-286, lines 311-312).

  • Line 87 - 88. Patients recruited to this study were given cisplatin, vp-16, vincristine, cyclophosphamide, and either carboplatin or Adriamycin. Etoposide was not one of the chemotherapeutic agents given to those patients. Why was Etoposide-resistant NB cell line used to identify miRNAs associated with chemoresistance to other chemo drugs? Have there been any published studies indicating that an NB cell line resistant to different cytotoxic chemotherapeutic agents would have the same miRNA expression profile? Also, chemoresistance can be intrinsic or acquired. Did those non-responders exhibit intrinsic resistance or acquired resistance to the treatment?

Apologies for this lack of precision - we used VP-16 (the commercial name of Etoposide) instead of Etoposide and so now we have changed it accordingly (line 88).

Notable, we used Etoposide-resistant NB cells to identify miRNAs associated with chemoresistance, since we previously demonstrated that these NB cells are multi-drug resistant (see reference in the text [12]).

Concerning the question as to whether the resistance can be acquired or intrinsic, we believe that it is intrinsic since the patients' samples were collected before starting with the drug treatment (see lines 86-87). We have added this consideration (lines 283-286).

  • Regarding the data shown in Table 2 and Table3, survival analysis should be conducted, and Kaplan-Meier survival curves for OS and PFS should be presented.

Regarding the data shown in Table 2, it concerns the genomic difference between the parental cells and Etoposide-resistant cells, thus no Kaplan-Meyer survival curves can be drawn. As for the data in Table 3, it is clear that we had analyzed 5 responder patients and 5 non-responder patients (see column Induction response yes and no), all of whom were either alive or dead, respectively. Therefore, the number of patients analyzed is low and the number of events (relapse and death) also so low that Kaplan-Meyer plots are intuitive. In addition, we did not find any significant difference in the expression of miRNAs between responder and no responder patients, except for miR-16, making a stratification of patients, according to miR16 levels, meaningless.

We look forward to hearing from you promptly and in the meantime,

Kind regards

Barbara Marengo

Barbara Marengo, PhD

Assistant Professor

University of Genoa

Department of Experimental Medicine

via L.B. Alberti 2

16132 Genova

tel. +390103538831/fax. +390103538836

Reviewer 2 Report

Minor

  1. Proofreading by an English speaker is mandatory. An example of colloquialisms includes metastatization (lines 69, 386). Typos and grammatical errors include a semi-colon doublet (line 88), marker”s” (line 264), “more higher” (line 39), etc…
  2. Figure 1 and 2 are the same data displayed differently. Furthermore, figure 2 is unacceptable as a figure. The y axis is not scaled. The description of the y-axis in the legend is incorrect. The green vertical line showing the threshold of log2 5-fold differential expression value should be located around 2.32. Finally, the direction of the differential expression on the x-axis is unknown.
  3. Please provide the list of all dysregulated miRNA in HTLA-ER.
  4. Can the patients in Table 4 be classified as responders or non-responders ?

Major

  1. mir218 is upregulated in MYCN-amplified NB and in metastatic NB (Table 1). However, the authors report the downregulation of mir218 in the etoposide-resistant MYCN-amplified HTLA cell line (Figure 3) and the downregulation of miR-218 at etastatic sites in patients (Figure 5). The discrepancy between the literature and the results of the present study needs to be explained.
  2. From the list of dysregulated miRNA in HTLA-ER, the authors only focussed on those already known to be dysregulated in NB from the literature including their own (reference 13, Table 1). What is the biological significance of the dysregulated miRNA in HTLA-ER that have not been previously described in neuroblastoma (Figures 1 and 2) ? The emphasis on downregulated miRNA is also not justified. What is the biological significance of the upregulated miRNA in HTLA-ER (Figures 1 and 2)?
  3. The only novel finding in this study is the potential association between the metastatic process in neuroblastoma development and the downregulation of miR-19b and miR-218. A mechanistic study is required to explain the role of miR-19b and miR-218 downregulation in the neuroblastoma metastatic process.
  4. The purification process of bone marrow samples to obtain metastatic neuroblastoma cells is provided in references 14 and 15. The protocol provided indicates that the purified cell population comprises more than 95% neuroblastoma cells. However, in the present study, the authors state “NB cells in BM samples ranged between 5 and 35%, making the normal hematopoietic cells prevalent, thus masking the miRNA down-regulation, occurring in NB cells” (line 306-308). If the neuroblastoma cell content after purification is only up to 35%, then miRNA analysis has been performed mainly on heamatopoietic cells. This would invalidate the study of miRNA expression at neuroblastoma metastatic sites, the only novel finding in this study.

Author Response

Genova, 20th January 2021

For the kind attention of Editor-in-Chief

Journal of Personalized Medicine

Dear Editor,

Firstly, we would like to thank you for having reviewed the paper and thank the Reviewers for their most welcome and helpful comments that have gone towards improving the quality of the manuscript which is now enclosed.

In this revision phase, we have tried as far as possible to satisfy the requests and comments by modifying the main text.

For greater clarity, the changes made to the original manuscript have been highlighted in the text as follows:

-           parts added in red;

-           parts removed in red but barred.

As you will see, we have addressed all the points raised by the Reviewers and amended the manuscript accordingly as follows: 

REVIEWER 2

Minor

  • Proofreading by an English speaker is mandatory. An example of colloquialisms includes metastatization (lines 69, 386). Typos and grammatical errors include a semi-colon doublet (line 88), marker”s” (line 264), “more higher” (line 39), etc…

As required by the Reviewer, the text of the paper has been checked by an English mother-tongue speaker and accordingly modified. At the same time, the text has also been revised in order to eliminate any typos or grammatical errors.

  • Figure 1 and 2 are the same data displayed differently. Furthermore, figure 2 is unacceptable as a figure. The y axis is not scaled. The description of the y-axis in the legend is incorrect. The green vertical line showing the threshold of log2 5-fold differential expression value should be located around 2.32. Finally, the direction of the differential expression on the x-axis is unknown.

In line with the Reviewer's comment, Figure 2 has been removed in the new version of the paper and, consequently the numbering of figures has been changed accordingly.

  • Please provide the list of all dysregulated miRNA in HTLA-ER.

As required by the Reviewer, a list of all dysregulated miRNAs in ER-HTLA has been reported as Supplementary material (S1).

  • Can the patients in Table 4 be classified as responders or non-responders ?

The patients in Table 4 could be classified as responders and non-responders but, in this set of experiments, we have not found any differences in the expression levels of miRNAs, except for miRNA-19b, -218 and -338 that were down-regulated in MYCN-amplified metastases by about 30% in comparison with MYCN-amplified tumors. 

Major

  • mir218 is upregulated in MYCN-amplified NB and in metastatic NB (Table 1). However, the authors report the downregulation of mir218 in the etoposide-resistant MYCN-amplified HTLA cell line (Figure 3) and the downregulation of miR-218 at metastatic sites in patients (Figure 5). The discrepancy between the literature and the results of the present study needs to be explained.

Regarding the expression of miRNA-218, it is necessary to clarify that miRNA-218 was found to be up-regulated in MYCN-amplified and metastatic NB and that it is not in contradiction with the down-regulation of miRNA-218, that we have found only after chronic Etoposide exposure of MYCN-amplified NB cells Therefore, we have clarified on page 7, lines 249-252 that this is the first time that miRNA-218 expression has been related to NB chemoresistance.

In our opinion, the discrepancy between the literature and the results of the present study could be related to the different types of samples analyzed. In order to better discuss the results (page10,  lines 346-351), the reference [70] ZeKa F et al. has been added

  • From the list of dysregulated miRNA in HTLA-ER, the authors only focused on those already known to be dysregulated in NB from the literature including their own (reference 13, Table 1). What is the biological significance of the dysregulated miRNA in HTLA-ER that have not been previously described in neuroblastoma (Figures 1 and 2) ? The emphasis on downregulated miRNA is also not justified. What is the biological significance of the upregulated miRNA in HTLA-ER (Figures 1 and 2)?
  • The only novel finding in this study is the potential association between the metastatic process in neuroblastoma development and the downregulation of miR-19b and miR-218. A mechanistic study is required to explain the role of miR-19b and miR-218 downregulation in the neuroblastoma metastatic process.

The emphasis on the down-regulated miRNAs is based on the fact that chemoresistance is likely to be associated with an increased expression of proteins that allow the cancer cells to survive in the presence of the drug, rather than associated with a decrease in the expression of the same proteins. Since miRNAs suppress the expression of mRNAs and thus reduce the expression of the corresponding proteins, we have concentrated our attention on the down-regulated miRNAs because their decrease corresponds to an increase in mRNAs and proteins related to tumor survival. Please note that the results of our experiments on tumors and metastases actually do not suggest that the metastatic process is dependent on the down-regulation of miR-19b and miR-218. In fact, we have never stated it. Our results simply indicate that these two miRNAs are less expressed in MYCN-amplified metastases than in the MYCN-amplified primary tumor cells whereas this difference was not present in the absence of MYCN amplification. Given the pivotal role of MYCN in regulating miRNA expression, it is likely that the difference between primary tumor cells and metastases does not depend on the metastatic process but on the level of MYCN mRNA in the two sets of samples. In fact, metastases were 95% pure whereas tumors had around 70% of tumor cells (lines 92-95).

Furthermore, since the present study is essentially focused to study the involvement of miRNAs in the development of chemoresistance, the title has been modified as follows: “Role of miRNAs in the acquisition of chemoresistance in neuroblastoma”.

  • The purification process of bone marrow samples to obtain metastatic neuroblastoma cells is provided in references 14 and 15. The protocol provided indicates that the purified cell population comprises more than 95% neuroblastoma cells. However, in the present study, the authors state “NB cells in BM samples ranged between 5 and 35%, making the normal hematopoietic cells prevalent, thus masking the miRNA down-regulation, occurring in NB cells” (line 306-308). If the neuroblastoma cell content after purification is only up to 35%, then miRNA analysis has been performed mainly on heamatopoietic cells. This would invalidate the study of miRNA expression at neuroblastoma metastatic sites, the only novel finding in this study.

Sincere apologies for not having explained clearly enough that there were actually two groups of patients whose samples were studied. Of the first group, the unselected bone marrow (BM) samples were analyzed and this is why we made the statement about purity. Of the second one, the primary tumors and the metastases were analyzed. In this latter case, as previously explained, the tumors had more than 70% of neoplastic cells and the metastases were 95% pure (lines 92-95). To avoid any misinterpretation, we have clarified this point better in the Materials and methods section and we have specified that the use of materials came from two groups of patients (lines 85-91).

Regarding the concern that having studied the unselected BM invalidates the novelty of our study, we do believe that this is the first time that the BM metastatic site microenvironment has been related to a potential deregulation of a miRNA involved in chemoresistance.

We look forward to hearing from you promptly and in the meantime,

Kind regards

Barbara Marengo

Barbara Marengo, PhD

Assistant Professor

University of Genoa

Department of Experimental Medicine

via L.B. Alberti 2

16132 Genova

tel. +390103538831/fax. +390103538836

Round 2

Reviewer 1 Report

The authors have addressed all of my concerns. 

Reviewer 2 Report

All minor comments were adressed adequately.

Only the first and fourth major comments were adressed adequately.

Major comment 2:

There is no adequate justification for the miRNA selection criteria used. Out of the 10 most significantly downregulated miRNA, only 2 (miR-26b and miR-338) have been selected for further analysis.

The authors provided the list of 35 dysregulated miRNA from their cell line screen in supplementary Table S1. 11 miRNA were selected for further analysis, including miR-19b. However, miR-19b is not amongst the most significantly dysregulated miRNA in supplementary Table S1. Similarly, the selection of miR-16-1 is not justified. It does not appear in Tables 1 or S1.

The authors argue that only miRNA downregulation is relevant to cancer because miRNA downregulation leads to overexpression of the target gene. This is not acceptable. Many miRNAs are considered oncogenes (i.e. their downregulation would impede tumour progression). If the target gene is a tumour suppressor gene, then miRNA downregulation would increase the tumour suppressor gene expression and therefore halt/slow tumour progression.

Major comment 3:

The association of miR-19b, miR-218 and miR-338 downregulation with metastasis still requires more experimentation to delineate a mechanism of action explaining the association.

Furthermore, in Figure 3, the authors show that miR-16-1 expression in 5 non-reponders is 80% of that in 5 responders. A 20% difference in an extremely limited number of patients is unlikely to have any clinical significance for diagnosis or therapy. Therefore, more experiments are also required to explain the association between miR-16-1 downregulation and chemoresistance.

Overall, the use of only 1 pair of cell lines for the in vitro study and only 20 patients for the clinical study is too weak to describe these miRNAs as potential diagnostic tools in neuroblastoma.